# Neural computations underlying inverse reinforcement learning in the human brain

Sven Collette[1,2]*, Wolfgang M Pauli[1,2], Peter Bossaerts[3,4], John O'Doherty[1,2]

[1]Division of Humanities and Social Sciences , California Institute of Technology, Pasadena, United States; [2] Computation and Neural Systems Program, California Institute of Technology, Pasadena, United States; [3]Florey Institute of Neuroscience and Mental Health, The University of Melbourne, Melbourne, Australia; [4]California Institute of Technology, Pasadena, United States

**Abstract** In inverse reinforcement learning an observer infers the reward distribution available for actions in the environment solely through observing the actions implemented by another agent. To address whether this computational process is implemented in the human brain, participants underwent fMRI while learning about slot machines yielding hidden preferred and non-preferred food outcomes with varying probabilities, through observing the repeated slot choices of agents with similar and dissimilar food preferences. Using formal model comparison, we found that participants implemented inverse RL as opposed to a simple imitation strategy, in which the actions of the other agent are copied instead of inferring the underlying reward structure of the decision problem. Our computational fMRI analysis revealed that anterior dorsomedial prefrontal cortex encoded inferences about action-values within the value space of the agent as opposed to that of the observer, demonstrating that inverse RL is an abstract cognitive process divorceable from the values and concerns of the observer him/herself.

DOI: https://doi.org/10.7554/eLife.29718.001

*For correspondence:
sven.collette@gmail.com

**Competing interests:** The authors declare that no competing interests exist.

## Introduction

When learning through observing others' actions, two major strategies have been proposed: in imitation learning, an individual simply learns about the actions of an observed agent, in essence by learning to copy the agent's behavior. An alternative strategy called inverse reinforcement-learning (inverse RL) is to use the agent's actions to infer the hidden or unobservable properties of the environment, such as the reward outcomes available for pursuing particular actions. Then the observer can use that knowledge to compute his/her own subjective expected value for taking distinct actions in the same environment. While in imitation learning, the observer learns nothing about the structure of the environment other than the action tendencies of the agents they observe, in inverse RL the observer has acquired knowledge of the world which becomes ultimately abstracted from the actions of the agent they observe. Although imitation learning can work well under a number of situations, particularly where the agent can be assumed to have similar preferences to the observer, inverse RL offers greater flexibility to learn from diverse agents, even under the situation where the agent has very distinct preferences or goals to that observer.

The computational framework of inverse RL was originally developed by researchers in the artificial intelligence community (**Ng and Russell, 2000**; **Abbeel and Ng, 2004**). Unlike in forward problems of reinforcement learning (RL) (**Sutton and Barto, 1998**), the link between action and reward is lost and the goal becomes to infer, from the agent's behavior and knowledge of the agent's preferences, relevant information about the reward function which can then be used in turn to maximize one's own reward.

In the realm of cognitive sciences, the application of computational modeling tools have surged as a means of gaining insight into social behavior. Observational learning has recently been formalized as an inverse problem, where observed behavior is used to infer the hidden structure at the root of that behavior under the assumption that observed choices are goal-directed (*Baker et al., 2009*, *Baker et al., 2011*; *Goodman et al., 2009*; *Shafto et al., 2012*). Recent computational modeling work has focused on how humans create an understanding of other's preferences and the similarity to one's own past choices, to guide future behavior (*Gershman et al., 2017*). Computing abstract social inferences has been characterized as emerging already in early infancy (*Lucas et al., 2009*; *Jara-Ettinger et al., 2016*). There is empirical evidence that subjective preferences emerge in toddlers as early as 16 to 18 months (*Repacholi and Gopnik, 1997*; *Ma and Xu, 2011*), and by the age of three children use inferences about social preferences to guide their own decisions (*Fawcett and Markson, 2010*).

Within the nascent sub-field of social computational neuroscience, a number of studies have utilized computational fMRI in order to gain insight into the neural mechanisms underlying observational learning. When learning through observation from an agent's actions and outcomes, evidence for a process called vicarious reinforcement learning (RL) has been found whereby reward prediction errors generated by observing the rewards obtained by another as opposed to being received by oneself have been reported in the striatum (*Burke et al., 2010*; *Cooper et al., 2012*). However, vicarious RL can only ever work under situations where the rewards being obtained by another agent can be directly observed. The problem focused on here is the situation in which the actions of the other agent can be observed, but the reward outcomes cannot.

Other studies have found evidence for the engagement of several brain structures when making inferences about the hidden intentions and traits of another agent. In particular, regions of the brain often referred to as the mentalizing network which includes the dorsomedial prefrontal cortex (dmPFC) have been implicated in this type of inference (*Frith and Frith, 2006*; *Hampton et al., 2008*; *Nicolle et al., 2012*), whereas posterior superior temporal sulcus (pSTS) and adjacent temporoparietal junction (TPJ) has been found to reflect learning signals about relevant attributes of other individuals' behavior (*Hampton et al., 2008*; *Behrens et al., 2009*; *Boorman et al., 2013*; *Dunne and O'Doherty, 2013*). Yet, these studies have not addressed the capacity to make inferences about the structure of the environment through observing an agent beyond mentalizing about the intentions/traits of that agent him/herself. This capacity is the essence of inverse RL that we investigate in the present study.

To address this issue, we implemented a novel functional magnetic resonance imaging (fMRI) experiment involving participants (always the observers) and social agents (the agents). After having learned about food preferences of two observable agents, one with similar and another one with dissimilar preferences to themselves, the participants observed these same agents perform on a slot machine task, in which the agents learned to obtain their preferred food outcomes. Importantly, participants never got to see the food outcomes themselves. In order to assess what participants had learned from observing the choices of the other agents, we asked participants to play the same slot machine task.

We hypothesized that regions of the brain traditionally implicated in mentalizing such as the TPJ/pSTS and dmPFC would be involved in implementing inverse RL, and in subsequently utilizing knowledge about the outcome distributions acquired through inverse RL to guide behavior. More specifically, building on previous findings of a key role for TPJ/pSTS in encoding of updating signals or prediction errors during social learning (*Hampton et al., 2008*; *Boorman et al., 2013*), we hypothesized that those same regions would play a role specifically in the acquisition and updating of information using inverse reinforcement learning through observing the behavior of others, while dmPFC was hypothesized to play a role in using the acquired knowledge to ultimately guide behavior.

## Results

### Observational slot machine learning paradigm

43 human participants underwent fMRI while performing an observational learning (OL) task (*Figure 1A*), in which on each of two sessions they observed an agent make repeated binary choices between pairs of slot machines, sampled by the computer from a pool of three slot machines.

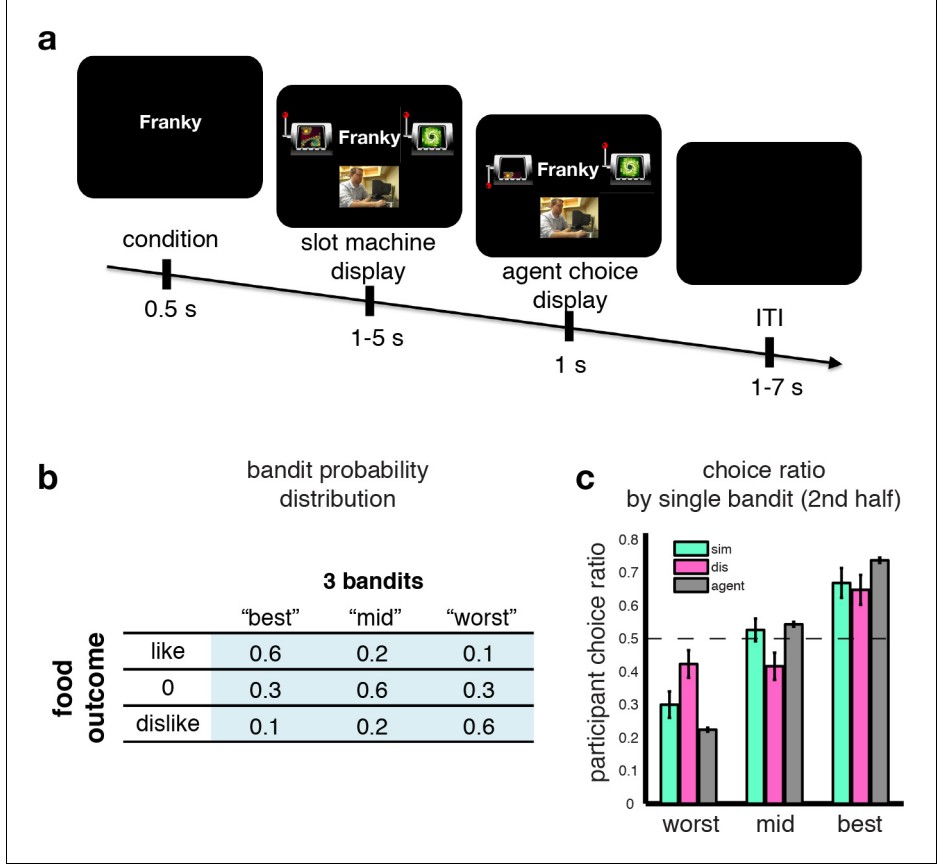

**Figure 1.** Observational slot machine task with hidden outcomes. (**A**) Trial timeline. The first screen signals whether the agent or the participant has to make a choice. Subsequently two slot machines are presented, along with on agent trials a (pseudo) video feed of the agent making a selection, and the choice of the slot machine is revealed after a jittered interval. (**B**) The probabilistic food distribution behind the slot machines: each of the three slot machines is labeled by a unique fractal and yields the same three food items, but at differing stationary probabilities, as indicated in the table. (**C**) Agent and participant choice ratio. Choice ratio is defined as choice frequency of a slot machine given its total number of presentations. Agent performance (grey bars) are collapsed across conditions, and depicted in agent-referential space. Participant performance (cyan for similar, pink for dissimilar) is depicted in self-referential space. Regardless of whether they are observing a similar or dissimilar agent, participants are equally good at choosing their best slot machine, nearly matching the agent's performance.

DOI: https://doi.org/10.7554/eLife.29718.002

The following figure supplement is available for figure 1:

**Figure supplement 1.** Additional task information.

DOI: https://doi.org/10.7554/eLife.29718.003

Participants were instructed that each of the slot machines would on each trial deliver one of three different food items as a prize. These food items were selected to differ in their subjective value to the participant (one was a preferred food item, another middle ranked, and another least preferred, drawn from an initial list of 10 food items that were earlier ranked for preference). These items were delivered with differing (but fixed across session) probabilities. The distributions varied across the machines, but participants were instructed that the underlying distribution of these food items was hidden from both the participants and the agents (*Figure 1B*). The critical feature of this paradigm is that participants never got to see any outcomes of the slot machines while participants understood that the agents did receive the outcomes. Occasionally (1/3 of trials), the participants themselves were presented with the same slot machines as those played by the agents, and they had to make a choice. The participants were instructed that one of their choice trials would be selected at random at the end of the experiment, that one of the food outcomes would be delivered to the participant

according to the probabilities assigned to that chosen slot machine and that they would be asked to consume outcomes. Thus, participants were motivated to ensure that they chose the slot machine with the highest probability of yielding a food outcome that they preferred as opposed to one that they liked the least. Crucially, in a previous preference learning phase (see Materials and methods for details), participants got to learn about the food preferences of the two agents they subsequently got to observe playing on the slot machines. One agent had similar food preferences to the participant while the other agent had dissimilar food preferences. Prior to the onset of the observational learning task, to confirm the efficacy of our preference learning procedure we tested participants' knowledge of the other agent's preferences. Each participant correctly identified the preference rankings for each agent over the foods, as well as correctly classifying the agents as similar or dissimilar to themselves (*Figure 1—figure supplement 1*).

## Behavioral results

Spearman's rank correlation coefficients between participants' choices in self-referential space and agent choices in agent-space, revealed that participants' choice ratios were generally in line with the agents': rs_sim = 0.7 (similar condition), rs_dis = 0.6 (dissimilar condition; see Materials and methods for details). These correlations also provide evidence that participants succeeded in learning to choose the slot machine with the highest likelihood of a preferred outcome.

Yet, learning from a dissimilar agent should be more challenging than learning from a similar agent since in the former case the participant should take into account the fact that the agent has different preferences to themselves and use this information to guide their choices, whereas in the similar condition the agent can either adopt a policy of simply imitating the agent or can easily infer that agent's preferences as they are similar to themselves. We therefore expected that participants would perform worse in the dissimilar condition compared to the similar condition. Indeed, participants performed significantly worse overall in the dissimilar condition compared to the similar (after Fisher transform: one-tailed p=0.04). We further compared the choice ratio of the slot machines across conditions on trials after the learning of the agent has reached a plateau (*Figure 1—figure supplement 1*). If participants used a simple imitative behavioral (and/or inverting) strategy, we should observe symmetrical performance changes for the best and worst machine in the dissimilar compared to the similar condition, whereas the mid machine should stay unchanged. In other words there should be an accuracy trade-off, that is, decreased choice frequency for the best machine while at the same time an increase of choices of the worst machine. Here we found that although participants performed equally well in choosing their 'best' slot machine in both similar and dissimilar conditions (t-test between similar and dissimilar choice ratios: P>0.6), the worst slot machine was picked more often in the dissimilar condition than in the similar (P<0.05) and the mid-ranking machine was chosen significantly less often in the dissimilar condition compared to the similar (P<0.05, *Figure 1C*). This non-symmetrical choice difference between observer and agent, that is, equal performance for the best, but not simply reversed for the mid and worst machines, alludes to a more sophisticated strategy than simple imitation behavior.

## Computational modeling of OL

To elucidate the computational mechanisms involved in solving the task we implemented an inverse RL algorithm in which participants infer the distribution of outcomes over the slot machines chosen by the agents, given knowledge about the agent's utility function (the learned preferences) and the observed actions (Materials and methods).

An alternative strategy is to deploy imitation RL, wherein the participant learns to predict the actions taken by the other agent, based solely on the history of actions taken by that agent. These predictions (which are formulated as value signals) are applied to the participants' own choices without any need to make an inference about the outcome distribution over the slot machines. In essence this algorithm would implement a pure imitation strategy: simply copying the actions of the opponent. Pure imitation would succeed in the similar case but fail abjectly in the dissimilar case, as the observer would end up choosing slot machines leading to least as opposed to most preferred outcomes. To give this imitation strategy a fighting chance to cope with learning from the dissimilar other, we modified the algorithm so that it also encoded as a free parameter a representation of how similar the agent's preferences were to oneself (which we could assume was learned before the

observational learning task). This similarity metric is used to invert the observer's value function when choosing from learning the actions of the dissimilar agent. This adapted form of imitation RL can in principle successfully learn from both similar and dissimilar agents, and hence we refer to this adapted model when referring to imitation RL in analyses and figures. We also constructed a different imitation RL model with a counterfactual learning component (cf-imitation RL) that also learned about the value of action not taken on a given trial, while incorporating separate learning rates for those two updating strategies (Methods and Materials for more details).

Yet another viable strategy for solving this task is for the observer to assume that the agents have distributions over the preference rankings for the different slot machines without considering the outcome distributions per se. A preference-learning algorithm can then simply update the observers' beliefs about the agent's machine preferences based on the observed behavior of the agents, and then explores the slot machines using a softmax procedure (Materials and methods for model details, and comparisons in *Figure 2—figure supplement 1*.

We tested which algorithm provides the best account of participants' behavior by using a Bayesian Model Selection (BMS) process (Materials and methods). Out of all tested models, inverse RL was found to outperform Imitation RL overall (*Figure 2A*) and in almost all individual subjects (*Figure 2B*). The model similarly also outperformed the other candidate algorithms (*Figure 2—figure supplement 1*). The choices estimated by the inverse RL model match participants' behavior very well when plotting model predictions against participants' actual (*Figure 2—figure supplement 1*).

To demonstrate that our two main competing models specify unique behavioral strategies as opposed to making similar predictions about behavior, we constructed confusion matrices, where each cell depicts the frequency with which each model wins based on behavior generated under each model, and inverted by itself and all other models. If there is a high probability that a model is confused with another, we should observe high exceedance probabilities in the off-diagonal cells. However this comparison framework shows that our main models are indeed not confused, and hence make different predictions (*Figure 2C*, and *Figure 2—figure supplement 1*).

Building on these initial model comparison results, we next aimed to elucidate the qualitative difference in the performance of the models. When comparing model choice predictions and participant choice ratios. While both imitation RL and inverse RL do quite reasonably in predicting choices for the most and least preferred slot machine, we found that the behavior that discriminates the model predictions the most is with regard to participants' behavior to the middle preferred slot machine when learning from the dissimilar other. There was considerable variation across individuals in the selection of the middle preferred slot machine: this variation was much more effectively captured by inverse RL compared to imitation RL (*Figure 2D*). A more detailed explanation for why inverse RL would be expected to perform better than imitation RL in capturing participants' choice behavior especially for the middle preferred slot is given in Materials and Methods.

## Neural representation of outcome prediction in agent-referential preference space

To determine whether participants' neural signals tracked value for the observed actions in self-referential or agent-referential preference space, we constructed two regressors with trial-by-trial value predictions for the actions taken by the similar and dissimilar other. To obtain an estimate of the value from the perspective of the agent, the agent's preferences over the outcomes were multiplied by the outcome probabilities estimated by inverse RL (cf. Materials and methods). We tested for regions correlating with these value signals across the whole brain. If value signals are represented in the observer's brain in the preference space of the other agents, then we would expect significant positive correlations with the observee's value signal for both the similar and dissimilar agents. On the other hand, if the value signals are represented with respect to the observer's own (self-referential) preferences and not those of the agent (agent-referential), then we would expect to find a significant positive correlation with the value regressor in the similar condition and a significant negative correlation with the value regressor in the dissimilar condition, because the dissimilar agent's preferences are opposite to those of the observer.

We found no evidence of a region exhibiting correlations for value signals in self-referential space anywhere in the brain. Instead, we found a significant cluster in the dmPFC (*Figure 3A*; peak x = 0, y = 40, z = 40, whole brain cluster corrected threshold of FWE at p<0.05 with a height threshold of

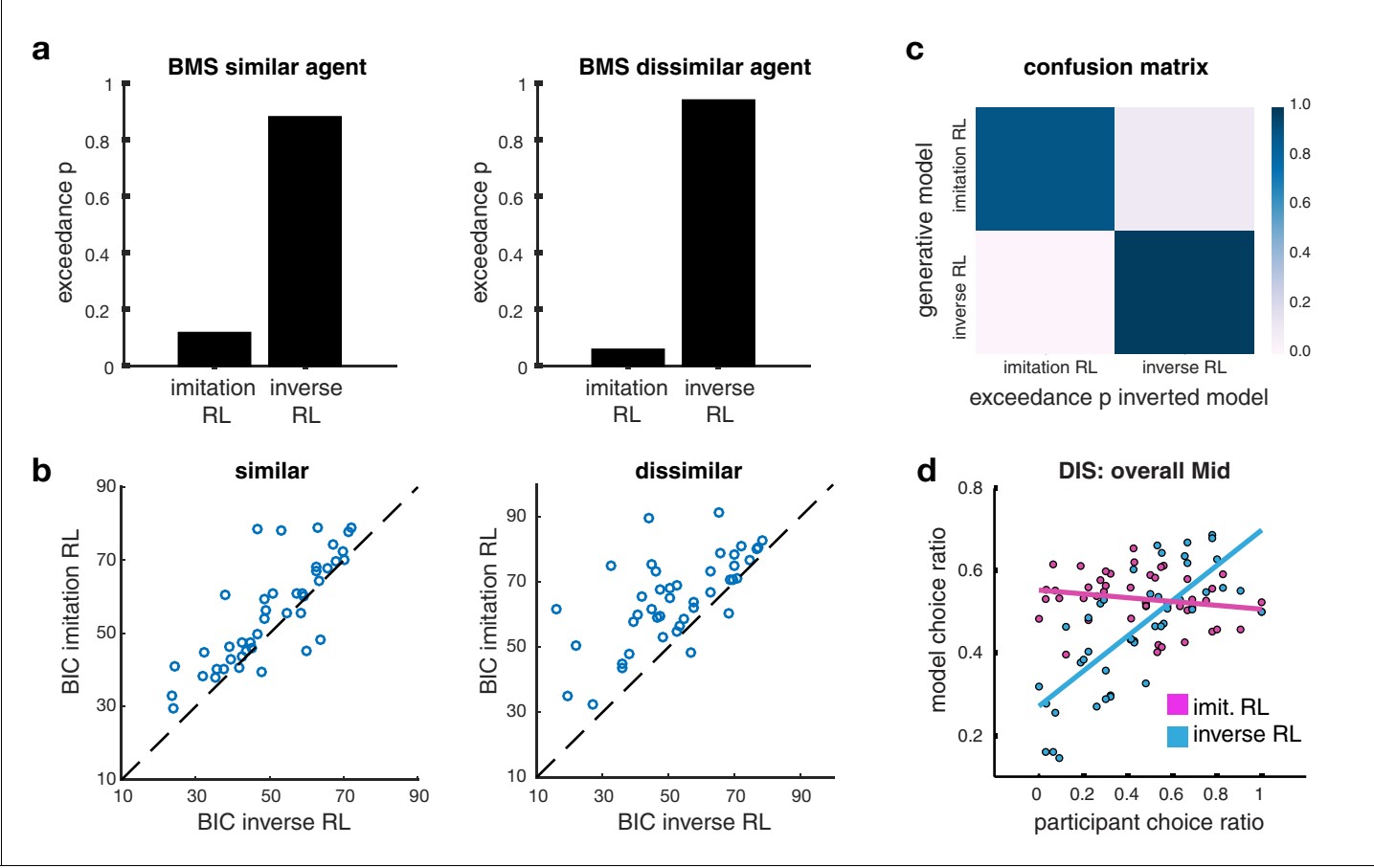

**Figure 2.** Model comparison. (A) Bar plots illustrating the results of the Bayesian Model Selection (BMS) for the two main model frameworks. The inverse RL algorithm performs best, across both conditions (similar and dissimilar). On the left, the plot depicts the BMS between the two models in the similar condition; on the right the plot shows the BMS in the dissimilar condition (BMS analysis of auxiliary models are shown in *Figure 2—figure supplement 1A*). (B) Scatter plots depicting direct model comparisons for all participants. The lower the Bayesian Information Criterion (BIC), the better the model performs, hence if a participant's point lies over the diagonal, the inverse RL explains the behavior better. The figure on the left illustrates the similar condition; the plot on the right depicts the dissimilar condition. (C) Confusion matrix of the two models to evaluate the performance of the BMS, in the dissimilar condition. Each square depicts the frequency with which each behavioral model wins based on data generated under each model and inverted by itself and all other models. The matrix illustrates that the two models are not 'confused', hence they capture different specific strategies. Confusion matrices of the similar condition and for auxiliary models are shown in *Figure 2—figure supplement 1C*) (D) Scatter plots depict the participant choice ratio of the mid slot machine plotted against the predictions of the inverse and imitation RL models.

DOI: https://doi.org/10.7554/eLife.29718.004

The following figure supplement is available for figure 2:

**Figure supplement 1.** Additional model information.

DOI: https://doi.org/10.7554/eLife.29718.005

p<0.001) correlating positively with the agent's value signals in the similar condition, as well as positively with the agent's value signals in the dissimilar condition (*Figure 3B*, parameter estimates extracted via a leave-one-out approach to avoid a non-independence bias [*Esterman et al., 2010*]). Further, no statistical difference was found between conditions (t-test p>0.5). These results indicate that expected value signals for outcomes are represented in agent-referential preference space.

Next, we aimed to test if inverse RL best explained the correlation within dmPFC, or if the imitation RL model also captured this activity. We therefore performed a Bayesian Model Selection (BMS) analysis to compare the model correlations within an anatomically defined dmPFC region where we would expect inverse RL related computations to be implemented (*Dunne and O'Doherty, 2013*; *Apps et al., 2016*). Our results show that the anterior part of the dorsomedial cluster is much more

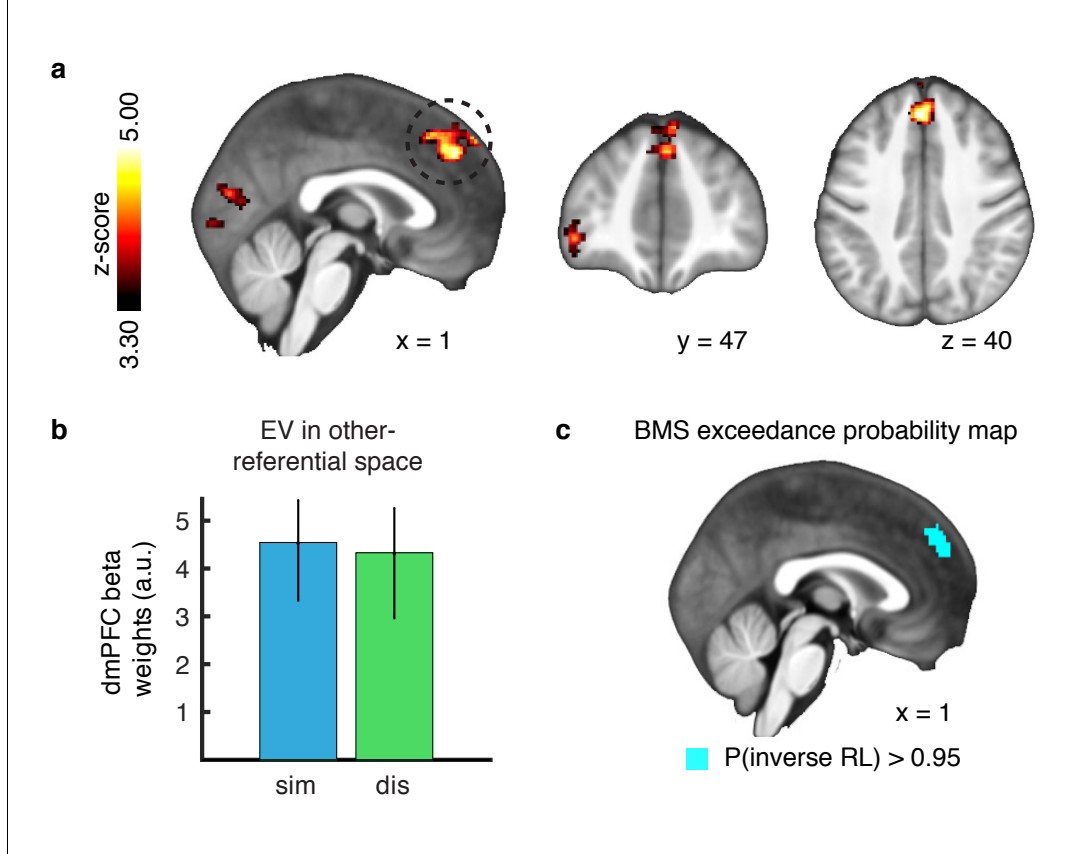

**Figure 3.** Outcome prediction signals in agent-referential preference space. (**A**) Neural response to parametric changes in inverse RL outcome prediction in agent-referential space. Activity in dmPFC at the time of presumptive agent decision significantly correlated with outcome prediction inferred from the inverse RL model, independent of condition. All depicted clusters survive whole brain cluster correction FWE at p<0.05, with a height threshold of p<0.001. Z-score map threshold as indicated, for illustrative purposes. (**B**) Effect sizes of the outcome prediction correlation in dmPFC cluster separately for each condition (similar = blue, dissimilar = green, parameter estimates are extracted with a leave-one-out procedure, mean ±SEM across participants, p<0.05 for both conditions). (**C**) Group-level exceedance probability map of the Bayesian Model Selection, comparing predictions from the imitation RL against inverse RL voxelwise in an anatomically defined dmPFC region. The depicted map was thresholded to show voxels where the exceedance probability for the inverse RL model is greater than p=0.95, revealing that the anterior part of dmPFC is much more likely to encode inverse RL computations.

DOI: https://doi.org/10.7554/eLife.29718.006

The following source data is available for figure 3:

**Source data 1.** areas exhibiting significant changes in BOLD associated with predicted outcome in similar and dissimilar.
DOI: https://doi.org/10.7554/eLife.29718.007

likely to be engaged in computing expectations derived from the inverse RL model, compared to the imitation RL (*Figure 3C*, exceedance probability threshold of p>0.95 for inverse RL).

## Inverse RL signals

We next tested for brain areas involved in updating beliefs in the inverse RL model at the time of feedback. We computed the KL divergence between posterior and prior outcome distributions for the chosen slot machine at the time of action observation and regressed the divergence signal against the BOLD response. We found significant correlations with the entropy reduction signal in both similar and dissimilar conditions, in TPJ/pSTS (whole brain cluster corrected threshold of FWE at p<0.05 with a height threshold of p<0.001). In addition we found significant correlations with this learning signal in several other brain regions including the pre-SMA and dorsal striatum (*Figure 4*). Furthermore no significant difference in the representation of updating signals was found between the similar and dissimilar conditions. Here, we compared the conditional beta values extracted using

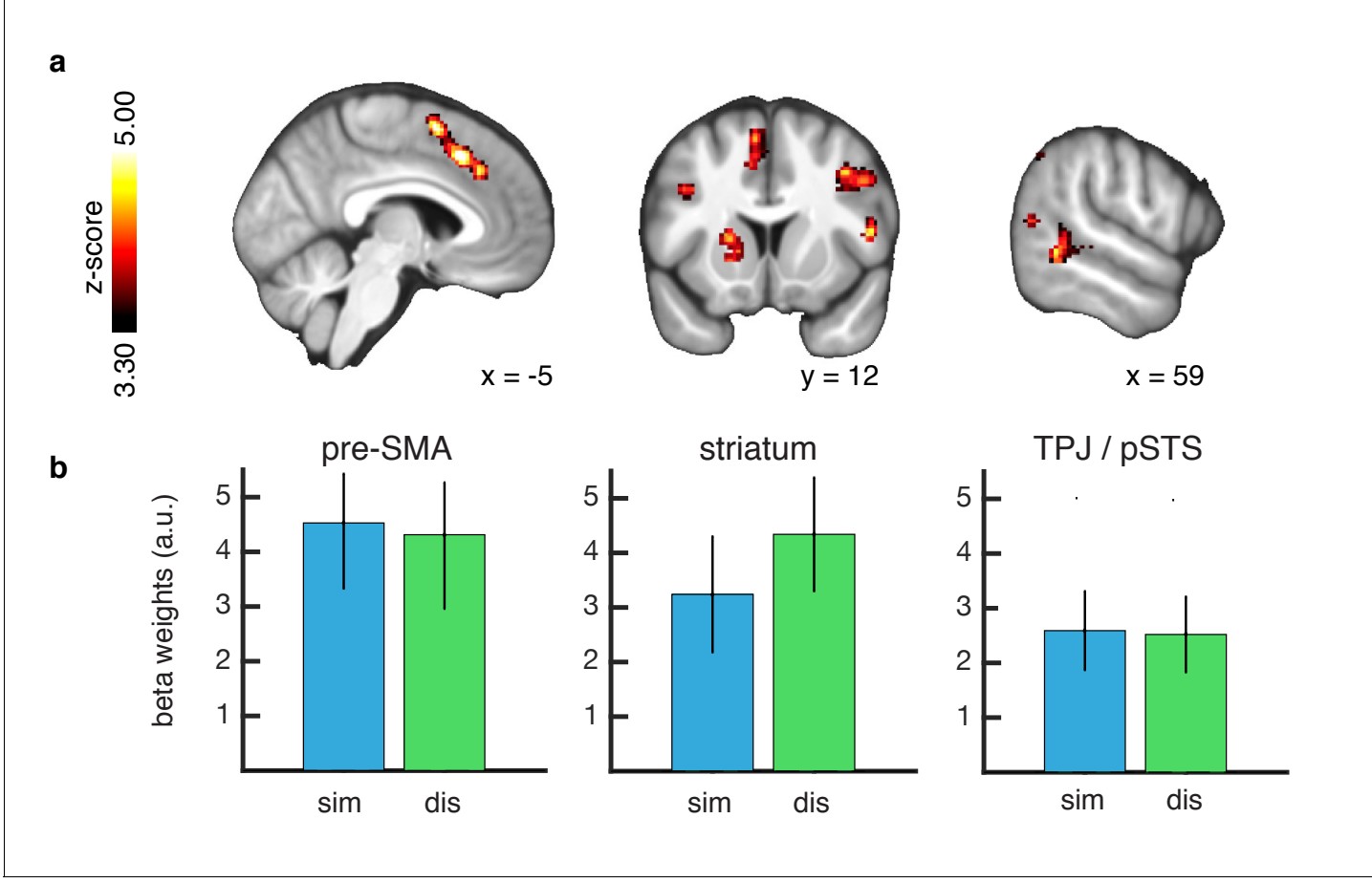

**Figure 4.** Learning signals during action feedback. (**A**) Z-statistic map of the inverse RL entropy signals during agent choice revelation is presented, relating to the update of the food distributions within the chosen slot machine. From left to right slices depict pre-SMA, striatum and TPJ/pSTS. All depicted clusters survive whole brain cluster correction FWE at p<0.05, with a height threshold of p<0.001. Maps are thresholded at Z-statistic values as indicated, for display purposes. (**B**) Effect sizes of the entropy correlations for all clusters, separately for each condition (similar = blue, dissimilar = green, L1O procedure, mean ±SEM across participants, all p < 0.05 for both conditions).
DOI: https://doi.org/10.7554/eLife.29718.008

The following source data is available for figure 4:

**Source data 1.** areas exhibiting significant changes in BOLD associated with entropy signals.
DOI: https://doi.org/10.7554/eLife.29718.009

a leave-one-out method as before. The findings suggest that learning from similar and dissimilar others involves overlapping computational mechanisms involving the same brain regions.

We also looked at the neural correlations with the prediction error signal from the imitation RL model in a separate GLM. However no cluster survived the predefined threshold (whole brain cluster corrected threshold of FWE at p<0.05 with a height threshold of p<0.001).

## Value correlations in dmPFC predict behavioral performance

To test whether BOLD responses in the brain while learning from the dissimilar agent is ultimately predictive of performance on the slot machine task, we extracted activity from the dmPFC region found to encode expected value in the agent space. We computed a behavioral index that measured the degree of success in integrating preference information about the dissimilar agent in order to guide personal choices, namely, the degree to which the participant matched the performance of the agent. We found a significant correlation between this social information integration index (SI index, Materials and methods) and the extent to which dmPFC represented learned values for the

dissimilar other (*Figure 5A*, p<0.01, R = 0.39). For comparison, we performed the same correlation against activity in the TPJ/pSTS, and we found no such significant correlation (*Figure 5B*).

## Discussion

Here, we implemented a novel fMRI experiment to shed light on how, at the computational and neural level, humans deploy knowledge about the preferences of other agents when learning about the environment through observing the actions of those agents. We designed our experiment in order to discriminate between two distinct strategies for implementing learning through observation. The simplest strategy is to learn about which action to take by imitating, or avoiding, the actions of the other agent. A more sophisticated strategy is to make an inference about the distribution of reward features available on the actions being pursued by the other agent and use this knowledge to guide one's own choices. We dissociated between these two strategies by exposing observers to three slot machines with hidden outcomes and two different agents, one who had similar preferences over available foods as the observer, and another one whose food preferences differed. While imitation could work effectively when learning from an agent with similar preferences, we hypothesized that such a strategy would not be effective enough when learning from observing the actions of an agent with dissimilar preferences.

Using this experimental design, we were able to show that participants used the more sophisticated strategy of inferring, from the agent's actions, the hidden distribution of food outcomes for each choice. This inverse RL algorithm provided a better account of the observer's behavior even if we altered the competing model, the imitation RL model, to accommodate diverging preferences between observer and agent. The inverse RL model continued to outperform the imitative learning model even if we added the capacity to learn from counterfactual actions. Finally, inverse RL also outperformed a slightly more sophisticated algorithm than imitation, in which the observer was assumed to learn the agent's ranking of the three available slot machines, without explicitly learning about the distribution of outcomes available on those actions. Taken together, these findings suggest that the human brain implements a mechanism akin to inverse reinforcement learning, and that the knowledge acquired through this mechanism is used to guide behavior.

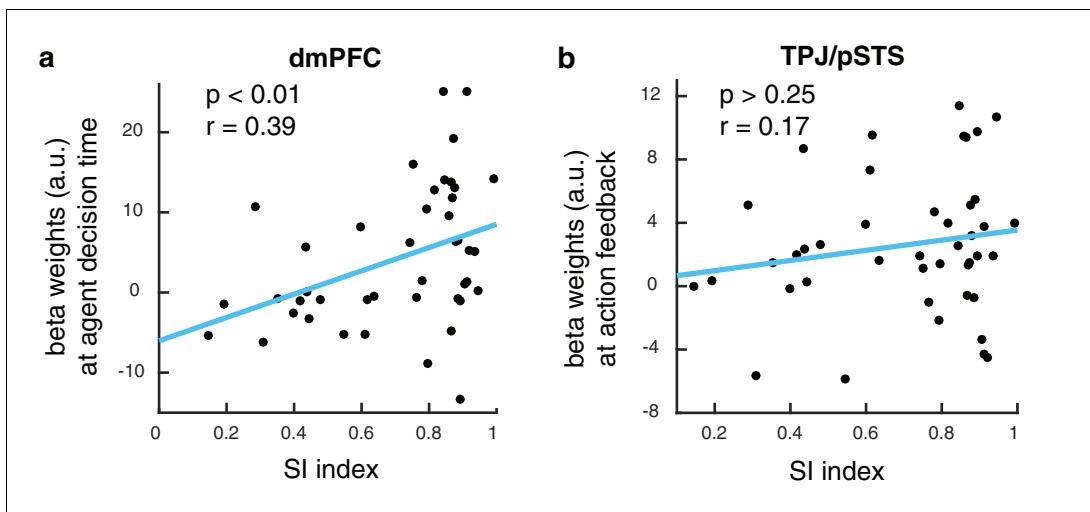

**Figure 5.** dmPFC signal predicts performance in slot machine game. (A) Scatter plot showing beta estimates of outcome prediction signals in the dmPFC ROI across participants, plotted against the social information integration index (SI index), which characterizes each participant's performance in the slot machine game. The higher the score, the better the participants are at inferring the best option for themselves from observing the other. (B) Scatter plot of entropy update signals in the TPJ/pSTS ROI across participants plotted against the social information integration index.
DOI: https://doi.org/10.7554/eLife.29718.010

Utilizing computational fMRI, we illuminated the brain systems involved in inverse RL. Two brain networks appear to be implicated: the mentalizing network, including TPJ/pSTS and dmPFC, as well as a network of brain regions implicated in goal-directed learning, including the anterior dorsal striatum and pre-SMA (*Kolling et al., 2012*; *Boorman et al., 2013*). The signals encoded in these areas associated with inverse RL go beyond prior reports about the involvement of regions of the mentalizing network and reward-learning network in observational learning. In previous studies implicating regions of the mentalizing network in social learning, the computational signals identified were about learning about the beliefs, traits and intentions of others, such as for example predictions about the other agent's beliefs in a competitive interaction (*Hampton et al., 2008*). Here, we focus on a quite distinct type of computation, the capacity to infer the distribution of rewards available on actions in the environment through observing the behavior of other agents. Furthermore, while previous studies such as Cooper et al. (*Cooper et al., 2012*) have found evidence for a role of the dorsal striatum in encoding reward prediction errors during observational learning, those signals were invoked when observing the rewards experienced by another agent i.e. vicarious reinforcement. Crucially in the present situation, the outcomes experienced are hidden away from the observer and must be inferred. Thus, the computations being revealed in the present are very distinct from those that have been reported previously in the literature.

Within the mentalizing network, the dmPFC was found to be involved in encoding the expected value of the actions chosen by the other agent, at the time that the agent's choice was being observed. The expected value of the action is computed using the observer's estimates of the distribution of outcomes over the chosen action. Importantly, there are two different ways in which the observer might represent expected values for the observed actions. One possibility would be to represent the value of actions in their own, observer-centric space, that is based on the value of subjective preferences the observer him/herself has for the food outcomes. Alternatively, the observer could represent the value of the food outcomes using acquired knowledge of the agent's preferences. Because observers were given the opportunity to learn from observing the actions of agents with similar and with dissimilar preferences, we were able to dissociate these alternatives. We found that the observer was representing the value of the observed actions in the agent's value space as opposed to their own. In other words, the observer was simulating 'what it would be like' to be the agent. This signal was represented in the dmPFC regardless of whether the agent was similar or dissimilar. These findings illustrate that the dmPFC can flexibly acquire knowledge about the outcome distributions through knowledge of the agent's preferences. The ability to do inference in agent-referential preference space, in contrast to self-referential, shows that participants approach social inference learning abstractly rather than concretely; in other words, the environment valuation variables are represented in the acting observee's perspective, in contrast to a simpler self-centered representation which would be directly relevant to their own choice situation.

Activity in an overlapping region of TPJ/pSTS was found to reflect the degree to which the observer updated his/her knowledge of the distribution of food outcomes on the different slot machines. Our findings build on previous studies using computational models to implicate TPJ/pSTS in updating predictions about other people's behavior, such as in the encoding of prediction errors about the influence one's action will have on the opponent's next game move (*Hampton et al., 2008*), prediction errors about others' intentions (*Suzuki et al., 2012*) or prediction errors about an agent's expertise (*Boorman et al., 2013*). The finding that this region is involved to a similar degree in updating the observer's knowledge for both similar and dissimilar social agents, and that it is not predictive of the observer's performance possibly suggests a more general role for this area in learning about hidden states in a social situation.

Beyond the mentalizing network, several other brain regions were found to be involved in learning about the outcome distributions, including the dorsal striatum, lateral prefrontal cortex and pre-SMA. Although we report signals referring to a reduction of entropy in these regions, it is important to note that, in our model, these variables are also related to value updates in accordance with previous findings, since the agent is concerned about the update of posterior mean values of the slot machines. Dorsal striatum has previously been implicated in learning about the value of actions, both in experiential learning tasks as well as in observational learning (*O'Doherty et al., 2004*; *Schönberg et al., 2007*; *Cooper et al., 2012*). Dorsolateral prefrontal cortex has previously been implicated in encoding representations of goal outcomes during decision-making (*McNamee et al., 2015*), as well as update signals of outcome estimates (*Barraclough et al., 2004*), implicating this

structure in goal-directed decision-making during experiential learning. Therefore, the fact that this region is involved in updating knowledge of outcome distributions that are inferred through observing others, is broadly consistent with a role for this region in representing and performing inference over outcomes whether those outcomes are being learned through direct experience or through observation. Finally, pre-SMA is a region that has previously been implicated in action-based decision making and in encoding the values of actions at the time of decision-making during experiential learning (*Seo and Lee, 2007*; *Wunderlich et al., 2011*; *Wunderlich et al., 2009*; *Kolling et al., 2012*; *Kolling et al., 2014*). Consequently, it appears that the capacity to acquire knowledge about outcome distributions through inverse RL depends on areas that may be specifically involved in social inference, as well as regions involved in action-based decision-making and experiential learning more generally.

We further found that the degree to which BOLD activity in the observer's dmPFC correlated with the values of the agent predicted the observer's success guiding their own behavior. We computed a numerical score (SI index, see Materials and methods for details) that captured the degree to which an individual observer was able to successfully incorporate knowledge of the dissimilar agent, and it exhibited a significant relationship with activation in dmPFC. These findings suggest that the more successfully an observer can represent the other agent's action-values in dmPFC, the better the observer can utilize these representations to guide their own behavior. These findings suggest a direct relationship between dmPFC activity and the capacity to integrate knowledge of other agents' preferences into the value computation, to effectively learn hidden outcome distributions.

At the time that the observer him/herself had to choose between the slot machines, we found evidence for involvement of a cluster in the medial prefrontal cortex (mPFC) positioned more rostrally to the region of dmPFC found to represent the agent's preferences. This rostromedial PFC sub-region exhibited signals correlating with the participant's value of the chosen option, specifically in the dissimilar condition. In contrast, at a more liberal statistical threshold we found chosen value signals in the ventral striatum only when observing the similar agent. These findings indicate that the observer is not only capable of representing expected values according to the subjective preferences of the observed agent, but that when the observer is required to make his/her own choices, the observer can use the same knowledge to compute expected values in own preference space, using a network of brain areas involved in social comparisons.

In conclusion, here we showed that the human brain is capable of performing a variation of inverse reinforcement learning, whereby one learns through observation of actions of other agents with potentially diverging preferences. By inferring the distribution of outcomes over available actions from the choices of the agent, participants successfully guided their own choices on the same task. We found evidence that a network of regions in the brain, especially areas traditionally implicated in mentalizing and theory of mind, such as the TPJ, pSTS and anterior dmPFC, appear to play a direct role in inverse RL. Importantly, neural signals in dmPFC implied that participants learned by representing action values in agent preference space, not their own, suggesting that their inference was abstract (i.e., disconnected from their own involvement) rather than concrete (i.e., directly relevant to their own choice situation). The findings are consistent with distinct contributions of the mentalizing network in different components of inverse RL. Posterior cortical areas such as the TPJ and adjacent pSTS were found to be especially involved in updating and learning about agent preferences and expectations irrespective of the specifics of the task participants had to fulfill themselves. By contrast, rostral dmPFC was particularly involved in representing agents' valuations in a way that these could eventually be used to guide own choices. Furthering an understanding of how the brain is capable of performing inverse RL may prove important in subsequent work aimed at identifying the nature of the dysfunction in certain psychiatric and neurological disorders involving deficits in social cognition such as in autism spectrum disorder.

## Materials and methods

### Participants

50 participants were recruited between the ages of 18 and 40 (mean: 27.1 years ± 4.9, 25 female). All were healthy, had normal/corrected to-normal vision, were free of psychiatric/neurological

conditions, and did not report taking any medications that might interfere with fMRI. All participants received $60 compensation, and were asked to remain for 30 min after the end of the MRI sessions, where they were allowed to consume the won food item ad libitum. We excluded seven participants due to excessive head motion or technical difficulties during scanning (leaving 23 male, 20 female). The included participants did not exhibit scan to scan movements of greater than one voxel within a session. The research was approved by the Caltech Institutional Review Board, and all participants provided informed consent prior to their participation.

## Experimental paradigm

The experiment is divided into three tasks, one outside the MRI scanner, and two inside. First participants revealed their preferences over 10 food items outside MRI. Second, inside the scanner, the participants learned the food preferences of an agent. Third, participants played an observational learning slot machine game with this agent in order to win a food item they could consume at the end of the experiment. The second and third phase is played once with a similar agent, and once with a dissimilar agent, pseudo-randomized across participants. A debrief after the experiment revealed that all participants thought they were observing real agents. Participants could deduce that both agents were male ('Franky' and 'Andrew' were used as names, as well as video-feeds of male agents). Experiments were programmed and displayed with Psychtoolbox within the Mathworks Matlab (RRID:SCR_001622) environment.

### Construction of food preferences of social agents

Before the MRI session, each participant was asked to express his/her food preferences over 10 food items through pairwise comparisons. The ranking we got from this pre-MRI phase was used to pair the participant with two agents: one very similar to her, and the other very dissimilar (*Figure 1— figure supplement 1*).

### MRI task 1

Preference learning. During the first MRI session, the participants learn the preferences of the agent through an observational learning game, and were subsequently asked to recall the agent's preferences and rate the similarity of their own preference to those of this agent on a scale from 1 to 7 (*Figure 1—figure supplement 1*). During this observational learning game, participants observed the agent as it expressed its preferences through pairwise comparisons, just as participants had done during the earlier construction of food preferences. The fMRI results from this phase of the experiment are not reported in the current manuscript due to space limitations and in order to maintain the focus of the present manuscript on inverse RL, but the results from this phase will be reported in a subsequent manuscript.

### MRI task 2

Observational learning of agents with diverse preferences. Following the preference learning of one agent, the participants played a slot machine game with that same agent. The participants observed the choices of the agent. Participants were not informed that the agent was artificial; rather they were instructed that they would be observing another player by live video-feed. Before the game starts, the participants see the three involved slot machines (depicted by different fractals) and the three food items that can be won. They were informed that each slot machine would deliver the three food items, but at different probabilities. However, they have no knowledge about these probabilities. The three food items are chosen from the five items ranked by the other agent: the most preferred, the middle and the least preferred. On each trial two out of the three slot machines were presented for choice. On 2/3 of the trials, participants observed the choice of the agent. Notably, the participants could not see the outcome of the chosen slot machine, however they knew that the agent would see the food outcome. On 1/3 of the trials, participants were asked to choose a slot machine, but here again no outcome was revealed. Importantly, this means that the participants never get to see any food outcomes – hence the choices were based solely on the observer's capacity to observe and learn from the actions taken by the other agents. This phase of the experiment totaled 150 trials: 100 observations and 50 self-choices. In this phase, the agent was implemented as a softmax learner, which was simulated for each session with a fixed learning rate and temperature.

The agent model started with flat priors over the three slot machines (e.g. 1/3 for each machine), it then made a softmax choice over the two displayed machines, and updated the value of the chosen slot machine through a weighted prediction-error, when the food outcome was revealed. In this simulation procedure, the food outcomes were numerically coded according to increasing preference, from 1 to 3. The outcomes were sampled stochastically, according to the fixed probability distributions of the chosen slot machine (cf. *Figure 3B*). Thus, the simulated agent exhibited the same behavioral characteristics in the similar and dissimilar condition.

## Computational models

### Inverse reinforcement learning (inverse RL): general ideas

Learning policies through a direct approach, that is, imitation or observational learning, where both agent and participant have the same goal, have recently been used in investigations of social phenomena (*Burke et al., 2010*). The aim of inverse RL is to implement an indirect learning approach, where a reward outcome distribution is recovered for which the agent's choices are optimal, i.e. which explains the agent's behavior (*Abbeel and Ng, 2004*; *Ng and Russell, 2000*). In our inverse RL approach the participant infers the reward outcome distribution of the slot machines using a framework with a participant-specific likelihood as to how the agent chooses given certain outcomes. Then, rather than mimicking the actions of the agent, the learned reward outcome distributions are subsequently used to construct participants' own optimal policies.

### Complete inverse RL model

The design of this task is such that the agent gets outcome information for each chosen action, while the participants only see the agent's (binary) choice without the outcome. Further, the instructions note that both the participant and the agent will each receive one actual food outcome at the end of the experiment, which is drawn randomly from all their respective outcomes during the game. Note that the approach described here does not guarantee the model to retrieve the true reward distribution, but to find a reward distribution which explains the actions of the agent, supposedly optimal, sufficiently well, relative to his preferences. Our inverse RL solution of the computational problem at hand proposes to incorporate information about the likelihood of choosing a machine given a certain outcome on the reward distribution. In this section we first describe a full model relying on true Bayesian updating, and then we propose a simplification of the updating, which is used in this study.

Agent. Let us denote $\Delta$ as the $3 \times 3$ matrix of outcome probabilities (rows: arms; columns: outcomes), and $v_a$ the vector of values of outcomes for agent ($3 \times 1$). Prior $p_t (\Delta|\theta)$ is Dirichlet with parameters $\theta$ represents the trial index. Expected value of arms in trial t is denoted $V_t$ ($3 \times 1$ vector). It equals:

$$V_t = \int \Delta v_a p_t(\Delta|\theta)d\theta = \left[ \int \Delta p_t(\Delta|\theta)d\theta \right] \cdot v_a \tag{1}$$

Choice is softmax (logit), so the chance that arm c is chosen when pair offered is (c,u) is:

$$f(c|pair(c,u)) = logit(V_t(c) - V_t(u)|\beta) \tag{2}$$

with u as unchosen option; $\beta$ the exploitation intensity parameter. We write the updating of $\theta$ based on outcome o after choice of arm c using Dirichlet updating equations as follows:

$$p_{t+1}(\Delta(c,\cdot)|o,c,\theta) \leftarrow p_t(\Delta(c,\cdot)|o,c,\theta) \tag{3}$$

where denotes the cth row of $\Delta$. With the Dirichlet distribution, there are no updates on beliefs about the rows of $\Delta$ of the unchosen arm or arm not offered. Let us assume the observer has the same prior as the agent (Dirichlet with same $\theta$), and that she knows this. (It is possible to allow the agent to have a different prior – different value of $\theta$ – but this would introduce additional complications).

The observer does not see outcomes; she only observes choices. Outcomes therefore constitute a sequence of hidden variables, like the 'state vector' in the Kalman filter. A prior distribution of outcome sequences can be computed however, and with the likelihood of choices given outcomes, the

observer can compute a posterior distribution of outcomes, as follows: The likelihood of any choice sequence given an outcome sequence can be obtained by fixing the outcome sequence, then combining the posterior given the outcome (*Equation 3*) and the value given the posterior (*Equation 1*) with the choice given the value (*Equation 2*). The prior of an outcome sequence can be obtained in the same way, but instead of taken outcomes as given, one draws outcomes trial-by-trial using the (trial) posterior belief of Δ. The problem is, of course, that these computations effectively mean that one is estimating hyper-dimensional integrals (over time, and over values of Δ). Once the observer has formed posterior beliefs of outcomes given choices, she can use those to determine posterior beliefs of Δ given the choice sequence. This way, she forms expectations of the evolution of agent beliefs over time, and because hers and the agent's are the same, of own beliefs (of Δ) as well. Because the complexity of these integrals renders this complete inverse RL model to be computationally intractable in practice, we implemented a reduced form of inverse RL that is capable of performing approximate inverse RL inferences with the advantage of being much more tractable and hence plausible as a model of human inverse RL inference, described below.

## Approximate inverse RL

A reduced-form version of the updating can be formulated, as follows: To determine choices, the agent (and observer) only uses posterior mean values of Δ (see *Equation 1*). So, the observer is only concerned about tracking the update of the posterior mean. This posterior mean is the same for the observer and agent, since they have the same priors. The observer knows that the agent is solving an exploration/exploitation problem, whereby the agent tends to choose more frequently the arms that he believes provide a higher chance of a more favorable outcome for him (*Equation 2*). Hence, if an arm is chosen, it is more likely that it generates the – for him – better outcome. If we list the outcomes in order of agent preferences so that elements of $v_a$ are decreasing, then it is reasonable for the agent to assume that the posterior mean probabilities of the chosen arm are updated as follows:

$$\overline{\Delta}(c,i) \leftarrow \epsilon_c(i)\overline{\Delta}(c,i)$$

where i (=1, 2, 3) indexes outcomes, and $0 > \epsilon 1, \epsilon 2, \epsilon 3 > 1$ denotes matrix element (c,i).

By the same token, the unchosen arm is likely to have less favorable chances of generating the good outcomes (from the point of view of the agent) and hence,

$$\Delta(c,i) \leftarrow \epsilon_u(i)\Delta(u,i)$$

where u identifies the unchosen option in the pair of arms offered to the agent, and the vector [$\epsilon 1$, $\epsilon 2$, $\epsilon 3$ ] is flipped. The outcome probabilities of the arm that was not offered in the trial is not updated.

Does it make sense to update beliefs of the row of Δ corresponding to the unchosen option? Notice that the Bayesian does not do so (see discussion below *Equation 3*). However, it is not always the case that the agent chooses the most preferred option; he sometimes explores and chooses a lesser-valued arm. This arm tends to produce the outcome with lower value (to him), and his beliefs will evolve accordingly: he will increase the chance of the low-value option. How can we capture this? The lesser-valued arm can be identified by the frequency with which he chooses this arm across trials: more often, it is the unchosen arm. Hence, to capture updates in beliefs about this lesser-valued arm, we increase the probability of lesser-valued outcomes of the unchosen arm in each trial. The updating constants $\epsilon_c(i)$ and $\epsilon_u(i)$ depend on the observer's (and agent's) prior beliefs of $\Delta(\theta)$ as well as the agent's exploitation intensity $\beta$. Since we don't know either, we estimate the updating constants from the data (observer and agent choices). On account of its tractability, this approximate inverse RL scheme (as opposed to the complete inverse RL described earlier) was implemented in the analyses reported in the current manuscript.

## Imitation RL model

A simpler alternative approach is an imitation learning scheme update using a prediction error rule as follows. The imitation value V of each of the three slot machines A,B,C are initialized at VA = VB = VC=0.5. On trials in which (for instance) the slot machine A is chosen, when the agent's choice is revealed, the update of the chosen slot machine uses a prediction error thus:

$$V_{obs\_A}(t+1) = V_{obs\_A}(t) + \eta \cdot (O_A - V_{obs\_A}(t))$$

$\eta$ is the learning rate, scaling the impact of the prediction error information. $O_A$ is the executed choice of slot A on trial t, which takes the value 1. To allow imitation learning more flexibility, we add an inversion parameter $\theta$ on self-choice trials, which describes the degree to which the observer inverts the expected values of slot machines before making a choice, based on their belief about the degree of similarity between the agent's preferences and themselves. Specifically, $V_{self\_A} = (1-\theta) \cdot V_{obs\_A} + \theta \cdot (1 - V_{obs\_A})$. For example, in the case of an observer learning from an agent with identical preferences the inversion parameter $\theta$ would be close to zero, and hence the observer relies on the same value function as learned from the agent, whereas if the agent is judged to have opposite preferences to the observer, then $\theta$ would be close to 1, and the observer will have an inverted value function for the slots compared to the observee. The observer's choice is then expressed as a soft-max function between the values for the slots, with a free parameter $\beta$ characterizing the stochasticity of the choices as a function of the value difference between the slots.

## Imitation learning with counterfactuals

For our counterfactual imitation learner (cf-Imitation RL) in addition to the update of the chosen slot machine, the value of the unchosen (but available) slot machine is updated too, following the same rule, with its own separate learning rate $\eta\_unchosen$ and setting Outcome unchosen = 0. Just as with the Imitation RL model, we added an inversion parameter $\theta$ on self-choice trials, which describes the degree to which the participants invert the action probabilities of slot machines before making a choice.

## Fitting procedure

We assumed that the likelihood of participants' self-choices is given by a softmax function over the value space of the two displayed slot machines. Parameters were fit using Matlab (RRID:SCR_001622) optimization under the constraint function fmincon to minimize the log-likelihood of the model fit with regards to the participants' own choices. This procedure was iterated with three randomly chosen starting points within the constraints to increase the likelihood of finding a global rather than local optimum. Note that the similarity is taken into account implicitly by the likelihoods of the inverse RL model; in our main imitation RL model we included a parameter to weigh the degree to which the observer intents to invert the expected values of slot machines before making a choice. For models including a decay factor, the fmincon fitting was additionally performed iteratively (because of the integer numbers of this parameter). Decay factor was fit with constraints [1 100] (i.e. denoting integer trial numbers), inverting parameter $\theta$ with [0 1], choice temperature $\beta$ (stochasticity) with [2 22].

## Why does inverse RL and imitation RL make different predictions about participants' choice behavior and why are those predictions especially distinct for choices of the middle preferred slot?

The agents that we constructed that the observer sees, are artificial agents and those agents have a linear preference function for the most, middle and least preferred foods. As a consequence (assuming a reasonable choice temperature), their behavior will be approximately linear – they will try to choose the best machine most often and the machine associated with the greatest probability of the lowest valued option least often. The middle machine will be approximately in the middle in terms of choice frequency on average. This of course depends on the objective probability distributions over the outcomes available on the given slots, that is, in a case where the middle ranking machine has probabilities much closer to the top ranking machine, the choice probabilities of the agent would end up such that the agent would choose the middle machine with a frequency closer to that of the best machine. But in the case of the present task design and parameters, it is the case that choice probabilities will be approximately linearly ordered over the machines. Now, the imitation RL will essentially learn to mirror those choice probabilities – as it learns solely through observing the choice tendencies of the agent. Let's consider instead inverse RL. This model is not concerned with learning directly from the choice behavior of the agent, instead it is concerned with inferring the outcome probabilities over the slots. As an observer, assuming I have learned sensibly about those

outcome distributions, I can then use my own preference function to guide my choices over the slots. For instance I might really like food A (my most preferred outcome), but I also might be quite partial to food B (my middle preferred outcome), and I might really hate food C (the least preferred outcome). The point is I don't need to have a linear preference function over the outcomes. My choice proportions over the slots can therefore more flexibly reflect my underlying preferences for the goods – I don't simply have to mirror the agent's preference function (or invert it). The free parameters in the inverse RL model can flexibly capture those differences in preferences. Although ostensibly they are used to update beliefs (i.e. as a likelihood function), these parameters can capture individual participants' flexible preferences over the goods.

Now why does this difference in model predictions between inverse and imitation RL show up especially for the middle preferred outcome and not the best and worst preferred? The reason is that the best slot is always going to be strongly dominated by the worst, unless people are truly indifferent between the best and worst foods which is not likely. Thus, both imitation and inverse RL should capture that well. However, the middle food might vary a lot more in its relative preference for participants and hence its likely that there is going to be more variance in choices for this across subjects. This is why inverse RL does better in this situation.

Note that inverse RL will be better equipped to generalize too across different kinds of agents — if the agents were not to have a linear preference function, inverse RL ought to more robust to this — and still enable the observer to learn about the slots in a way that allows the observer to flexibly make their own choices based on their own preference functions.

Furthermore, inverse RL will also be better able to generalize under situations where new slot machines are introduced — if for example I see the same (or even a different) agent make choices over other unique slot machines in a completely new context — inverse RL will be able to guide choices under situations where those other slots are now presented in pairs with the original slot machines — because the observer can use knowledge of the outcome distributions to compute expected values on the fly for those slots in the novel pairings. Imitation RL would be hopelessly lost here because it's value signals are relative to the other options available during the learning phase i.e. they are cached values analogous to model-free RL in the experiential domain (*Daw et al., 2005*).

## SI index

We computed a social information integration index (SI index) to get a general measure of the similarities between the agent's and the participants' choice behavior during phase 2. We therefore normalized the distribution of choice ratios for the three slot machines and calculated the entropy between both distributions. For easier understanding we inverted the magnitudes, and hence an SI closer to one indicates very similar choice ratio behavior; for example in the dissimilar condition an SI close to one suggests a good performance of the participant, whereas an SI of 0 would mean that in the dissimilar case the participant mimics the agent, and hence his/her performance would be bad. Note that when correlating the SI post-hoc against slot machine choice ratios, we find a strong correlation between the SI and the mid versus worst ratio, i.e. how often the participant chose mid over worst when these two slot machines were presented.

## Neuroimaging data acquisition

The fMRI images were collected using a 3T Siemens Trio scanner located at the Caltech Brain Imaging Center (Pasadena, CA) with a 32-channel radio frequency coil. The BOLD signal was measured using a one-shot T2*-weighted echo planar imaging sequence. Forty-four axial slices were acquired in oblique orientation of 30 degrees to the anterior commissure–posterior commissure line, with a repetition time (TR) of 2780 ms, TE of 30 ms, 80° flip angle, 3 mm isotropic resolution, 192 mm $\times$ 192 mm field of view. A high-resolution T1-weighted anatomical image (magnetization-prepared rapid-acquisition gradient echo sequence, $1 \times 1 \times 1$ mm voxels) was acquired at the end of the session.

## fMRI data analysis

All image analyses were performed using SPM12 (rev. 6906; http://www.fil.ion.ucl.ac.uk/spm, RRID: SCR_007037), following a standard preprocessing pipeline. EPI images were realigned and

realignment parameters were included in subsequent GLMs. Each subject's T1 image was segmented into gray matter, white matter, and cerebrospinal fluid, and the segmentation parameters were used to warp the T1 image to the SPM Montreal Neurological Institute (MNI) template using SPM's DARTEL procedure. The resulting normalization parameters were then applied to the functional data in 2 mm isotropic resolution. Finally, the normalized images were spatially smoothed using an isotropic 6 mm full-width half-maximum Gaussian kernel. Whole brain analyses where performed by defining a general linear model (GLM) for each participant, which contained parametric regressors representing the computational variables at the slot machine onset (outcome prediction in agent-referential space for both options, confidence), at the time the agent's choice was revealed (update entropy), and at the time of participant choice (outcome prediction for chosen and unchosen slot machine, reaction time). The fMRI results depicted in the main figures are based on an uncorrected threshold of p=0.001 (which is illustrated as z>3.3 in the figures) combined with an FWE-corrected cluster threshold of p=0.05 (here, k = 114). All regression estimates from *Figure 3* and *Figure 4* were conducted based on a leave-one-out procedure (*Kriegeskorte et al., 2009*) to avoid non-independence bias. Specifically we ran a leave-one-subject-out GLM analysis on the group level (*Esterman et al., 2010*), and each GLM defined the cluster for the subject left out. For the Bayesian Model Selection (BMS) analyses during action feedback, we defined a dmPFC region of interest in the Harvard/Oxford atlas (label 103 restricting the ROI to cover only medial areas, i. e. $+12 > X > -12$). The definition of this ROI was motivated by two recent reviews (*Dunne and O'Doherty, 2013*; *Apps et al., 2016*) which each identified the key dmPFC region involved in social inference. For this analysis we used the first-level Bayesian estimation procedure in SPM12 to compute a voxelwise whole-brain log-model evidence map for every subject and each model (*Penny et al., 2007*; *Rosa et al., 2010*). Both GLMs differed only in one single regressor: the parametric modulator for the prior expectations. Then, to model inference at the group level, we applied a random effects approach at every voxel of the log evidence data falling within the same masks previously used for beta extractions, constructing an exceedance posterior probability (EPP) maps for each model. The imaging results are displayed on a pool-specific structural template created by DARTEL procedure on the anatomical images of all the participants, transformed to MNI space.

## Data availability

The full anonymized dataset from this study is available in the NDAR data repository https://ndar.nih.gov/ under the collection ID 2417. Summary information on the data (e.g. additional details about the experiment such as picture files or exact timings of stimuli) is available on the NDA homepage without the need for an NDA account. To request access to detailed human subjects data, you must be sponsored by an NIH recognized institution with a Federalwide Assurance and have a research related need to access NDA data. Further information as to how to request access can be found here https://ndar.nih.gov/access.html. The fMRI activation maps are available at neurovault (http://neurovault.org/collections/ZZHNHAJU/).

## Acknowledgements

This work was supported by the NIMH Caltech Conte Center for the Neurobiology of Social Decision Making (JPO). We thank Tim Armstrong and Lynn K Paul for support with the participant recruitment, and Ralph E Lee and Julian M Tyszka for assistance with the experiments.

## Additional information

### Funding

| Funder | Grant reference number | Author |
| --- | --- | --- |
| NIMH Caltech Conte Center for the Neurobiology of Social Decision Making | P50MH094258 | John O'Doherty |

The funders had no role in study design, data collection and interpretation, or the decision to submit the work for publication.

## Author contributions

Sven Collette, Conceptualization, Data curation, Software, Formal analysis, Validation, Investigation, Visualization, Methodology, Writing—original draft, Writing—review and editing; Wolfgang M Pauli, Investigation, Writing—review and editing; Peter Bossaerts, Resources, Software, Methodology, Writing—original draft, Writing—review and editing; John O'Doherty, Conceptualization, Supervision, Funding acquisition, Methodology, Writing—original draft, Writing—review and editing

## Author ORCIDs

Sven Collette (iD) http://orcid.org/0000-0002-0234-1867

## Ethics

Human subjects: Informed consent, and consent to publish was obtained from all participants prior to their participation. The research was approved by the Caltech Institutional Review Board.

## Decision letter and Author response

Decision letter https://doi.org/10.7554/eLife.29718.016
Author response https://doi.org/10.7554/eLife.29718.017

## Additional files

### Supplementary files

• Transparent reporting form
DOI: https://doi.org/10.7554/eLife.29718.011

### Major datasets

The following datasets were generated:

| Author(s) | Year | Dataset title | Dataset URL | Database, license, and accessibility information |
|---|---|---|---|---|
| Sven Collette, Wolfgang M Pauli, Peter Bossaerts, John O'Doherty | 2017 | Raw data for Neural computations underlying inverse reinforcement learning in the human brain | https://ndar.nih.gov/edit_collection.html?id=2417 | Accession no. 2417. Please see data availability statement for further access details. |
| Collette S, Pauli WM, Bossaerts P, O'Doherty JP | 2017 | Statistical fMRI maps for Neural computations underlying inverse reinforcement learning in the human brain | http://neurovault.org/collections/ZZHNHAJU/ | The fMRI activation maps are available at neurovault (accession no: ZZHNHAJU) |

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
