## [Decision Letter]

Thank you for submitting your article "Neural computations underlying inverse reinforcement learning in the human brain" for consideration by *eLife*. Your article has been reviewed by three peer reviewers, one of whom is a member of our Board of Reviewing Editors and the evaluation has been overseen by Timothy Behrens as the Senior Editor. The following individuals involved in review of your submission has agreed to reveal his identity: Samuel J Gershman (Reviewer #1).

The reviewers have discussed the reviews with one another and the Reviewing Editor has drafted this decision to help you prepare a revised submission.

Summary:

How people learn what actions lead to preferable outcomes is an important question at the intersection of learning and social cognition. This paper presents a behavioral, computational and neural characterization of observational learning in humans, using a paradigm that contrasts an "inverse reinforcement learning" account with an imitation account. The results suggest that people use inverse RL, tracking value in the observed agents value space rather than their own. This function is supported by regions of the brain involved in goal-directed learning, as well as mentalizing.

Essential revisions:

Though the reviewers differed on the value of the work for a pure behavioral or cognitive science audience, the reviewers with expertise in neuroscience were in agreement regarding the novelty and value of these findings for that domain. Nevertheless, the reviewers identified several points that would need to be addressed in a revision. These points largely cluster under three major themes and so are summarized accordingly below.

1) The framing and background need to be fully reworked in order to better situate the present study in the context of the prior cognitive science literature on observational learning and inverse reinforcement learning. Here are some specific examples from the reviewers of relevant literature that was not addressed sufficiently in the report.

a) Several other papers have also pursued this topic and should be integrated. Baker, Saxe and Tenenbaum, (2009), in an influential paper, showed how a computational theory of goal inference can make very strong and accurate predictions about behavioral judgments, and this was later followed up in many different ways by Jara-Ettinger, Noah Goodman, and others.

b) Considerable research on preference learning by children likewise pursues this topic.

Fawcett and Markson, (2010).

Lucas et al., (2009).

Lucas, C. G., Griffiths, T. L., Xu, F., Fawcett, C., Gopnik, A., Kushnir, T., Markson, L., & Hu, J. (2014). The Child as Econometrician: A Rational Model of Preference Understanding in Children. PloS ONE 9(3): e92160 doi:10.1371/journal.pone.0092160.

Ma and Xu, (2011).

Repacholi and Gopnik (1997). c) There is a large literature on social influence that the paper doesn't touch upon at all.

d) There is recent work by Gershman, Pouncy and Gweon, (2017) on social learning that may be relevant.

2) Several of the specific analyses lacked sufficient justification and so either appeared inappropriate or arbitrary.

a) Much depends on the adequacy of the small volume correction (SVC). Though SVC is an acceptable approach, it is challenging to do in a way that does not introduce hidden degrees of freedom or that is unrealistically specific about the a priori hypotheses. The methods state that SVC was applied to 12mm spheres based on peak voxels from prior work from the authors' lab. This is likely only a modest correction to threshold given the size of these spheres, so it is important to justify such highly specific and small regions. Likewise, the particular choice of prior papers appears arbitrary in the absence of a justification. Thus, it must be clear that these are uniquely justified, were chosen prior seeing the data, and would truly be the only activity in these areas that would have been expected a priori. Indeed, it is not clear such specific predictions can be justified, given that this is the first fMRI study of inverse RL and regions like dmPFC are larger than 12mm. A more reasonable alternative would be a volume encompassing the full anatomically-defined region where observed activation would have been considered consistent with a priori predictions.

b) It was hard to track what thresholds were used throughout the results, discussion, and figures. For example, the discussion mentions a number of regions beyond dmPFC, like DLPFC, striatum, and TPJ. Were these also SVC based on a priori regions? If so, how many a priori regions were included? If not, how were these corrected? The discussion also mentions regions that were evident at a more lenient statistical threshold, though it was not clear what threshold that was.

c) It was not clear from the description how the model-based regressors from the inverse RL versus imitation RL were included in the GLM and then used for model selection, making the specifics of the model-based fMRI approach difficult to evaluate. Were they included in separate models or in the same model in the standard SPM way (which enters them ordered in something like a hierarchical regression)? Or were they allowed to compete for variance? Were they correlated? These points are important to clarify in order evaluate this aspect of the methods.

d) A one-tailed test was applied for some behavioral tests but not others. Though it is understood that the authors might have a directional prediction justifying the use of a one-tailed test, one would have to make an argument why there is a directional prediction for these particular analyses and not for others. The chief concern is that the one-tailed test was chosen after the two-tailed test failed to be significant.

e) The text notes that participants were removed for movement exceeding 10mm. This seems like a lax movement threshold. Movement more than a voxel is difficult to correct, and presumably voxel sizes were smaller than 10mm. How many participants moved more than a voxel?

f) It was not clear where the effect sizes appearing the figures come from. Was that based on unbiased ROIs?

3) Certain aspects of the theoretical models and their predictions were difficult to evaluate based on the description in the text.

a) There are several missing equations Materials and methods section.

b) Results section: "If participants used a simple imitative behavioral (and/or inverting) strategy, we should observe symmetrical performance changes (accuracy decreases) for the best and worst machine, compared to the similar condition, whereas the mid machine should stay unchanged." This prediction is difficult to understand: why is it the case that simple imitation would not produce changes in the mid machine?

c) It would be useful to see a bar graph like Figure 1 for the key models. The relevant data is shown in Figure 2—figure supplement 1 but it's hard to directly compare this to the human data.

d) It is not clear why the models make different predictions for the middle preferred slot machine, or how the models take into account similarity between the observer and agent.

---

## [Author Response]

Essential revisions:Though the reviewers differed on the value of the work for a pure behavioral or cognitive science audience, the reviewers with expertise in neuroscience were in agreement regarding the novelty and value of these findings for that domain. Nevertheless, the reviewers identified several points that would need to be addressed in a revision. These points largely cluster under three major themes and so are summarized accordingly below.1) The framing and background need to be fully reworked in order to better situate the present study in the context of the prior cognitive science literature on observational learning and inverse reinforcement learning. Here are some specific examples from the reviewers of relevant literature that was not addressed sufficiently in the report.a) Several other papers have also pursued this topic and should be integrated. Baker, Saxe and Tenenbaum, (2009), in an influential paper, showed how a computational theory of goal inference can make very strong and accurate predictions about behavioral judgments, and this was later followed up in many different ways by Jara-Ettinger, Noah Goodman, and others.

We thank the reviewers for their helpful comments and their literature suggestions. We have reshaped the Introduction to underline the importance of the computational insights from the field of cognitive science, as follows:

[…] In the realm of cognitive sciences, the application of computational modeling tools have surged as a means of gaining insight into social behavior. Observational learning has recently been formalized as an inverse problem, where observed behavior is used to infer the hidden structure at the root of that behavior under the assumption that observed choices are goal-directed (Baker, Saxe and Tenenbaum, 2009, 2011; Goodman, Baker and Tenenbaum, 2009; Shafto, Goodman and Frank, 2012). Recent computational modeling work has focused on how humans create an understanding of other’s preferences and the similarity to one’s own past choices, to guide future behavior (Gershman, Pouncy and Gweon, 2017). Computing abstract social inferences has been characterized as emerging already in early infancy (Lucas, Griffiths and Fawcett, 2009; Jara-Ettinger et al., 2016). There is empirical evidence that subjective preferences emerge in toddlers as early as 16 to 18 months (Repacholi and Gopnik, 1997; Ma and Xu, 2011), and by the age of three children use inferences about social preferences to guide their own decisions (Fawcett and Markson, 2010).

Within the nascent sub-field of social computational neuroscience, a number of studies have utilized computational fMRI in order to gain insight into the neural mechanisms underlying observational learning. […]

b) Considerable research on preference learning by children likewise pursues this topic.Fawcett and Markson, (2010).Lucas et al., (2009).Lucas, C. G., Griffiths, T. L., Xu, F., Fawcett, C., Gopnik, A., Kushnir, T., Markson, L., & Hu, J. (2014). The Child as Econometrician: A Rational Model of Preference Understanding in Children. PloS ONE 9(3): e92160 doi:10.1371/journal.pone.0092160.Ma and Xu, (2011).Repacholi and Gopnik (1997).

Our Introduction has been adapted to reflect the important research on the roots of social learning in early infancy:

[…] Computing abstract social inferences has been characterized as emerging already in early infancy (Lucas, Griffiths and Fawcett, 2009; Jara-Ettinger et al., 2016). There is empirical evidence that subjective preferences emerge in toddlers as early as 16 to 18 months (Repacholi and Gopnik, 1997; Ma and Xu, 2011), and by the age of three children use inferences about social preferences to guide their own decisions (Fawcett and Markson, 2010). […]

c) There is a large literature on social influence that the paper doesn't touch upon at all.

In the present work, our goal was to investigate how computational processes underlying reverse inferences in a social setting are implemented in the brain. Traditionally social influence has been referred to changing one’s behavior to match the responses of other individuals or groups, and (Deutsch, M. and Gerard, H. B. (1955) ‘A study of normative and informational social influences upon individual judgment.’, The Journal of Abnormal and Social Psychology, 51(3), pp. 629– 636. doi: 10.1037/h0046408.) proposed an important distinction between normative and informational social influence. The latter is characterized by a desire for accurate understanding of the environment and “correct” behavior, and hence our work might be interpreted loosely within this context (as mentioned in (Gershman, Pouncy and Gweon, 2017), cited in our Introduction). However, we feel that, given this interpretation of social influence, and how it might be interpreted as conformity, we are barely touching this large field of research and hence a deeper dive into this literature might not be justifiable for our Introduction.

d) There is recent work by Gershman, Pouncy and Gweon, (2017) on social learning that may be relevant.

We thank the reviewers for suggesting the recent fine modeling work by Gershman, Pouncy & Gweon (2017), and we adapted our Introduction to reflect the importance of this study:

[…] Recent computational modeling work has focused on how humans create an understanding of other’s preferences and the similarity to one’s own past choices, to guide future behavior (Gershman, Pouncy and Gweon, 2017). […]

2) Several of the specific analyses lacked sufficient justification and so either appeared inappropriate or arbitrary.a) Much depends on the adequacy of the small volume correction (SVC). Though SVC is an acceptable approach, it is challenging to do in a way that does not introduce hidden degrees of freedom or that is unrealistically specific about the a priori hypotheses. The methods state that SVC was applied to 12mm spheres based on peak voxels from prior work from the authors' lab. This is likely only a modest correction to threshold given the size of these spheres, so it is important to justify such highly specific and small regions. Likewise, the particular choice of prior papers appears arbitrary in the absence of a justification. Thus, it must be clear that these are uniquely justified, were chosen prior seeing the data, and would truly be the only activity in these areas that would have been expected a priori. Indeed, it is not clear such specific predictions can be justified, given that this is the first fMRI study of inverse RL and regions like dmPFC are larger than 12mm. A more reasonable alternative would be a volume encompassing the full anatomically-defined region where observed activation would have been considered consistent with a priori predictions.

In addition to surviving SVC as originally reported in the manuscript, in fact, each all of the key results reported in the manuscript shown in Figure 3 and Figure 4 survive a whole brain cluster corrected threshold of FWE at p<0.05 with a height threshold of p<0.001 in SPM (where k=114). This thresholding approach is known to be robust against false positives (Woo, Krishnan and Wager, 2014; Flandin, G. and Friston, K. J. (2016) ‘Analysis of family-wise error rates in statistical parametric mapping using random field theory’, pp. 1–4. doi: 10.1073/pnas.1602413113.). In addition, the parameter estimate extractions were performed using a leave-one-out procedure to overcome a cluster selection bias.

We have now updated the text to report this one single whole brain corrected threshold throughout, and we now drop the SVCs, given the reviewer’s concern about possible arbitrariness of the selection criteria for the co-ordinates. The leave-one-out procedure for extraction of the parameter estimates is also now described in more detail in the manuscript.

We further defined anatomical regions of interest for the purpose of the Bayesian model comparison described under point **c** below. For this we utilized the Harvard/Oxford atlas to define the region of dmPFC where we would expect inverse RL related computations to be implemented. Motivated by two reviews outlining the role of this region in social computations, one by our group (Dunne and O’Doherty, 2013), and another by (Apps, Rushworth and Chang, 2016), we selected region 103 in the Harvard/Oxford atlas, restricting the ROI to cover only medial areas (i.e. +12 > X > -12).

The thresholding used throughout and details of the anatomical ROI definition are given in the text.

b) It was hard to track what thresholds were used throughout the results, discussion, and figures. For example, the discussion mentions a number of regions beyond dmPFC, like DLPFC, striatum, and TPJ. Were these also SVC based on a priori regions? If so, how many a priori regions were included? If not, how were these corrected? The discussion also mentions regions that were evident at a more lenient statistical threshold, though it was not clear what threshold that was.

We now make it clear throughout that all of the key results described in the manuscript survive whole brain FWE cluster correction. Regarding the regions described as being at a more “lenient” statistical threshold: since these results did not reach corrected significance, we have now excised any specific reference to these results from the revised manuscript for the sake of clarity. Nevertheless, these uncorrected findings are still reported in a supplementary table for future reference by interested researchers.

c) It was not clear from the description how the model-based regressors from the inverse RL versus imitation RL were included in the GLM and then used for model selection, making the specifics of the model-based fMRI approach difficult to evaluate. Were they included in separate models or in the same model in the standard SPM way (which enters them ordered in something like a hierarchical regression)? Or were they allowed to compete for variance? Were they correlated? These points are important to clarify in order evaluate this aspect of the methods.

Specifically, we used the first-level Bayesian estimation procedure in SPM12 to compute a voxelwise whole-brain log-model evidence map for every subject and each model (Penny, Flandin and Trujillo-Barreto, 2007; Rosa et al., 2010). Both GLMs differed only in one single regressor: the parametric modulator for the prior expectations. Then, to model inference at the group level, we applied a random effects approach at every voxel of the log evidence data falling within the same masks previously used for beta extractions, constructing an exceedance posterior probability (EPP) maps for each model. In Figure 3 we depict voxels with an EPP > 0.95 for the inverse RL model, which shows that the anterior part of the dorsomedial cluster is much more likely to be engaged in computing expectations derived from the inverse RL model, compared to the imitation RL. To avoid any possible circularity in the modal comparison procedure, we now implement this BMS within an anatomical ROI for dmPFC as described in our response to section a.

d) A one-tailed test was applied for some behavioral tests but not others. Though it is understood that the authors might have a directional prediction justifying the use of a one-tailed test, one would have to make an argument why there is a directional prediction for these particular analyses and not for others. The chief concern is that the one-tailed test was chosen after the two-tailed test failed to be significant.

We reported a one-tailed test at p = 0.04 showing that participants performed more poorly in the dissimilar condition. This one-tailed test was well motivated because we expected that participants would perform WORSE in the dissimilar condition compared to the similar condition. This is quite a natural hypothesis since learning from others who have dissimilar preferences ought to be more difficult than learning from a participant with similar preferences to oneself. We have now more clearly described our hypothesis in this regard, and better justified the use of a one-tailed test.

e) The text notes that participants were removed for movement exceeding 10mm. This seems like a lax movement threshold. Movement more than a voxel is difficult to correct, and presumably voxel sizes were smaller than 10mm. How many participants moved more than a voxel?

We agree this was an overly lenient motion correction threshold. However, out of all the participants who were included in the study, no individual exhibited scan to scan movements of greater than one voxel within a session. This is now clarified in the text.

f) It was not clear where the effect sizes appearing the figures come from. Was that based on unbiased ROIs?

We updated our figure text to make it clearer where the effect sizes came from. First, the results depicted in Figure 3 and Figure 4 survive whole brain correction using FWE cluster correction (p=0.001 voxel threshold, and p=0.05 cluster significance threshold (k=114)), which has shown to be robust against false positives (Woo, Krishnan and Wager, 2014; Flandin and Friston, 2016). Second, for the parameter extraction results, we extracted the parameter values from the clusters surviving the cluster correction for both condition using a leave-one-out procedure.

3) Certain aspects of the theoretical models and their predictions were difficult to evaluate based on the description in the text.a) There are several missing equations Materials and methods section.

The equations were not printed in text due to a LateX compilation error during the manuscript submission process. We apologize for failing to notice this. They are now visible in the text. Example: […] Expected value of arms in trial *t* is denoted *V_t_* (3 X 1 vector). It equals:

Vt=∫∆vapt(∆|θ)dθ=[∫∆pt(∆|θ)dθ]va. (1)

Choice is softmax (logit), so the chance that arm c is chosen when pair offered is (*c,u*) is:

f(c|pair(c,u))=logit(Vt(c)–Vt(u)|β) (2)

with *u* as unchosen option; β the exploitation intensity parameter. We write the updating of θ based on outcome *o* after choice of arm *c* using Dirichlet updating equations as follows:

Pt+1(Δ(c,∙)|o,c;θ)←pt(∆(c,∙)|o,c;θ). (3)

where c,∙ denotes the *c*th row of Δ.[…]

b) Results section: "If participants used a simple imitative behavioral (and/or inverting) strategy, we should observe symmetrical performance changes (accuracy decreases) for the best and worst machine, compared to the similar condition, whereas the mid machine should stay unchanged." This prediction is difficult to understand: why is it the case that simple imitation would not produce changes in the mid machine?

The agents that we constructed that the observer sees are artificial agents and those agents have a linear preference function for the most, middle and least preferred foods. As a consequence (assuming a reasonable choice temperature), their behavior will be approximately linear – they will try to choose the best machine most often and the machine associated with the greatest probability of the lowest valued option least often. The middle machine will be approximately in the middle in terms of choice frequency on average. This of course depends on the objective probability distributions over the outcomes available on the given slots, i.e. in a case where the middle ranking machine has probabilities much closer to the top ranking machine, the choice probabilities of the agent would end up such that the agent would choose the middle machine with a frequency closer to that of the best machine. But in the case of the present task design and parameters, it is the case that choice probabilities will be approximately linearly ordered over the machines.

Now, the imitation RL will essentially learn to mirror those choice probabilities – as it learns solely through observing the choice tendencies of the agent.

Let’s consider instead inverse RL. This model is not concerned with learning directly from the choice behavior of the agent, instead it is concerned with inferring the outcome probabilities over the slots. As an observer, assuming I have learned sensibly about those outcome distributions, I can then use my own preference function to guide my choices over the slots. For instance, I might really like Cheetos (my most preferred outcome), but I also might be quite partial to Lays Chips (my middle preferred outcome), and I might really hate boiled pigs feet (the least preferred outcome). The point is I don’t need to have a linear preference function over the outcomes. My choice proportions over the slots can therefore more flexibly reflect my underlying preferences for the goods – I don’t simply have to mirror the agent’s preference function (or invert it). The free parameters, epsilon 1,2,3 in the inverse RL model can flexibly capture those differences in preferences. Although ostensibly they are used to update beliefs (i.e. as a likelihood function), these parameters can capture individual participants' flexible preferences over the goods.

Now why does this difference in model predictions between inverse and inverse RL show up especially for the middle-preferred outcome and not the best and worst preferred? The reason is that the best slot is always going to be strongly dominated by the worst, unless people are truly indifferent between the best and worst foods which is not likely. Thus, both imitation and inverse RL should capture that well. However, the middle food might vary a lot more in its relative preference for participants and hence it is likely that there is going to be more variance in choices for this across subjects. This is why inverse RL does better in this situation.

Note that inverse RL will be better equipped to generalize too across different kinds of agents — if the agents were not to have a linear preference function, inverse RL ought to more robust to this — and still enable the observer to learn about the slots in a way that allows the observer to flexibly make their own choices based on their own preference functions.

Furthermore, inverse RL will also be better able to generalize under situations where new slot machines are introduced — if for example I see the same (or even a different) agent make choices over other unique slot machines in a completely new context — inverse RL will be able to guide choices under situations where those other slots are now presented in pairs with the original slot machines — because the observer can use knowledge of the outcome distributions to compute expected values on the fly for those slots in the novel pairings. Imitation RL would be hopelessly lost here because it’s value signals are relative to the other options available during the learning phase i.e. they are cached values analogous to model-free RL in the experiential domain.

Second, on a computational level, to test whether the models make indeed different qualitative predictions, we added evidence that the models are not confused. One way of doing this is to construct a confusion matrix. The idea behind this is that it should be possible for a model to recover the parameters from behavioral data that was generated from the model itself, i.e. it explains the self-generated data better than any other model. In the confusion matrix, the diagonal elements indicate the likelihood of a better fit on the self-generated data, while the off-diagonals represent the likelihood for other models.

Passages from the Results section have been updated for clarity and we have now added a completely new section to the Material and methods, which provides the intuitive explanation given here for why the models make different predictions about choice behavior (especially of the middle-preferred option).

c) It would be useful to see a bar graph like Figure 1 for the key models. The relevant data is shown in Figure 2—figure supplement 1 but it's hard to directly compare this to the human data.

We have updated Figure 2—supplementary figure 1B for better comparison with human data. This new figure uses the participant data as shown in Figure 2 and shows the choice probability for our main models, i.e. inverse RL and imitation RL. The bar graphs illustrate that the inverse RL is in average closer to the human choice behavior.

d) It is not clear why the models make different predictions for the middle preferred slot machine, or how the models take into account similarity between the observer and agent.

Please see our response to comment (b) above which addressed the concern about why the choices of the middle preferred machine best discriminates between the model predictions.

In relation to the similarity between the observer and the agent, please see the following which was added into the Material and methods section:

“The degree of similarity between the agent and observer is taken into account implicitly by the fitted likelihoods of the inverse RL model; in our main imitation RL model we included a parameter to set the degree to which the observer intends to invert the expected values of slot machines before making a choice. […]”